# Effect of Surface Mechanical Attrition Treatment on Torsional Fatigue Properties of a 7075 Aluminum Alloy

**Yizhuo Li** [1,*]**, Delphine Retraint** [2] **, Pengfei Gao** [2]**, Hongqian Xue** [3]**, Tao Gao** [3] **and Zhidan Sun** [2]

1   Xi'an Modern Chemistry Research Institute, Xi'an 710065, China
2   LASMIS, University of Technology of Troyes (UTT), 10004 Troyes, France; delphine.retraint@utt.fr (D.R.); pengfei.gao@utt.fr (P.G.); zhidan.sun@utt.fr (Z.S.)
3   School of Mechanical Engineering, Northwestern Polytechnical University, Xi'an 710072, China; xuedang@nwpu.edu.cn (H.X.); gtao@mail.nwpu.edu.cn (T.G.)
*   Correspondence: liyizhuo1990@gmail.com; Tel.: +86-15809222624

**Abstract:** The effect of Surface Mechanical Attrition Treatment (SMAT) on torsional fatigue properties of a 7075 aluminum alloy was investigated. A number of fatigue samples were heat treated to increase the sensitivity of the material to SMAT. Compared with the as-machined (AM) samples, the fatigue lives of their SMATed counterparts (AM-SMAT) tested under torsional loading increased under high stress amplitudes, but decreased under low amplitudes. However, the fatigue lives of heated and SMATed samples (HT-SMAT) increased under all the investigated stress amplitudes, compared with those that were heat treated (HT). It was also revealed that the cracking mechanisms are different for the samples in different states, and they are dependent on the imposed stress levels. The results show that SMAT could have both beneficial and detrimental effects on the fatigue lives depending on the testing conditions. The roles played by various factors, including residual stresses, grain refinement, and surface roughness, were analyzed and discussed to interpret the results.

**Keywords:** torsional fatigue; aluminum alloy; SMAT; fatigue life; fracture mechanism

## 1. Introduction

High-strength aluminum alloys are widely used in the aircraft industry due to their high strength-to-density ratio. Strengthened aluminum alloys, such as those of the 7xxx series, exhibit, despite high tensile strength, a relatively low fatigue endurance in high cycle fatigue (HCF) [1]. Therefore, extensive studies have been conducted to understand the fatigue behavior of high strength aluminum alloys over the past decades. Most of these studies were performed under uniaxial loading [2–5], but little data could be obtained for torsional fatigue. However, with the increase in structural design requirements and the improvement in testing systems, torsional fatigue behavior of structures is gaining increasing attention, but always constitutes a challenging research subject. In contrast to tension–compression fatigue tests, for which stress is rather homogeneously applied on the cross section, the distribution of the applied stress for torsional fatigue tests decreases gradually from the exterior to the interior of the specimen, which leads to different behaviors of materials under torsional fatigue [6]. A considerable number of theoretical and experimental studies have been conducted to examine the effect of loading parameters on torsional fatigue failures, and some significant achievements have been made [7].

Numerous techniques were developed or adapted to improve the mechanical properties of metals and alloys by introducing a gradient ultra-fine grained (UFG) structure at the top surface of materials. These techniques include, in particular, surface mechanical treatment, such as ultrasonic shot peening [8–10], sever shot peening [11–13], laser shock peening [14,15], deep-rolling [16], and roller-burnishing [17]. Among these different variants of the surface mechanical treatments, Surface Mechanical Attrition Treatment (SMAT) is one of the methods widely studied [18]. During SMAT, the surface of a material

is repeatedly impacted by multi-directional spherical shot with high kinematic energy, leading to a surface nanocrystallization (SNC) of the material [19]. However, the interior of the material is not mechanically affected, and the bulk material's characteristics, as well as its properties, remain unchanged [20,21]. Compared with coarse grained (CG) metals, SNC metals exhibit enhanced surface hardness and attractive mechanical properties, such as superior yield strength [22], as well as excellent fatigue properties [23,24].

It is well documented that although the beneficial factors, such as compressive residual stress [23,25] and grain refinement [26] introduced by SMAT, are capable of increasing fatigue strength, some detrimental factors, including rough surface [25,27–29], micro-cracks [30], and plastic strain localization [24,31] resulting from SMAT could also decrease fatigue endurance to some extent. Some authors mention the harmful effects of SMAT on fatigue strength for several materials under certain fatigue loading conditions [2,21]. It is well documented that both high strength and sufficient ductility are important to obtain high fatigue resistance for nanostructured metals and alloys. Unfortunately, the ductility of materials is, in general, severely impaired after severe plastic deformation treatment processes [3]. Therefore, it is necessary to improve the mechanical properties in terms of ductility and toughness [32]. It is well known that ductility can be obtained through heat treatment [33,34]. The combination of heat treatments with surface mechanical treatments can, thus, be performed to reinforce the fatigue strength of materials.

In this work, the effects of SMAT on fatigue properties of a 7075 aluminum alloy (AA7075) in as-received and heat-treated states are investigated under torsional loading.

## 2. Material and Experimental Procedure

### 2.1. Material

The material studied in this work is a commercial AA7075 subjected to a T6 heat treatment. The nominal chemical composition (in wt.%) of the alloy is given in Table 1. The received bars were machined into fatigue testing samples in order to investigate the effect of SMAT on the fatigue properties of this alloy. The samples after machining, i.e., in T6 heat treatment, are named 'as-machined' (AM) samples. The dimensions and the shape of the samples are shown in Figure 1. The machined fatigue samples had a dumbbell shape with a total length of 116.6 mm. The central reduced section had a minimum diameter of 4 mm. The dimensions and the shape of the samples were determined by referring to the literature [7,35,36] and by taking into account other factors (possibility of being treated by SMAT, capacity of the fatigue machine). It should be noted that two flats were machined on each head of the samples and flat grips were used to avoid gliding while imposing torsional loading.

**Table 1.** Chemical composition of the studied AA7075 alloy (in wt.%).

| Cr | Cu | Fe | Mg | Mn | Si | Ti | Zn | Al |
|---|---|---|---|---|---|---|---|---|
| 0.18–0.28 | 1.2–2.0 | 0.50 | 2.1–2.9 | 0.30 | 0.40 | 0.20 | 5.1–6.1 | Balance |

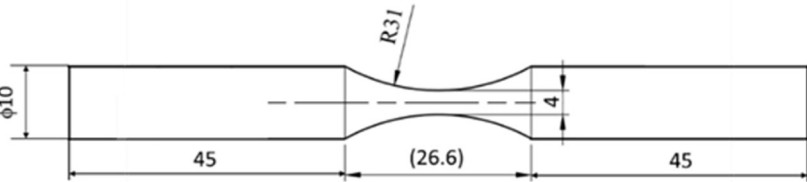

**Figure 1.** Shape and dimensions (in mm) of the samples used for torsional fatigue tests.

### 2.2. Heat Treatment

To increase the sensitivity of the material to SMAT, some of the fatigue samples were subjected to a specific heat treatment which included solution heat treatment and subsequent aging. This heat treatment process aimed to increase the size of the precipitates,

and consequently, to increase the ductility of the alloy [37–40]. Considering the compromise between the ductility and the mechanical strength of the alloy obtained after heat treatment, the heat treatment conditions adopted in this work were as follows: the temperature was increased to 450 °C and kept constant for 4 h, followed by a water quenching; aging was then carried out at a temperature of 185 °C for 12 h. The samples treated by heat treatment were named 'heat-treated' (HT) samples.

### 2.3. SMAT Process

A number of fatigue samples including both AM (as-machined) and HT (heat-treated) were then processed by SMAT. A SMAT set-up is based on the vibration of balls boosted by a high frequency (20 kHz) ultrasonic generator (Branson S.A.S., Rungis, France). The schematic description of the SMAT set-up can be found in the literature [41,42]. The central region of cylindrical fatigue samples is in an enclosed chamber and repeatedly impacted by high kinematic energy balls projected in multiple directions. In order to homogenously impact the cylindrical surface of the sample, an electric motor is used to keep the sample rotating during the SMAT process. The rotating speed used in this work was 30 rpm. A mass of 20 g steel balls with 2 mm diameter was used for the SMAT process. The distance between the surface of the sonotrode and the sample was 18 mm. In this work, according to the coverage assessment performed for a sample, a duration of 150 s was regarded as 100% coverage of ball impacts. Therefore, a duration of 25 min corresponding to a coverage of 1000% was chosen. According to the literature for various SMATed materials, this treatment should be able to generate a grain-size gradient with a superficial nanostructured layer at the treated surface [43–45], as well as a compressive residual stress field [18].

Both AM and HT samples were treated under the same conditions of SMAT, and the treated samples were named 'AM-SMAT' and 'HT-SMAT', respectively. Note that the roughness of the samples at these states was measured and several representative values are presented as follows: arithmetic mean roughness $R_a$ = 0.45 µm, total height of the roughness profile $R_t$ = 4.75 µm for non-SMATed samples (AM and HT), and $R_a$ = 0.25 µm and $R_t$ = 6.17 µm for SMATed samples (AM-SMAT and HT-SMAT).

### 2.4. Microhardness

To prepare a specimen for microhardness measurement, the central part of a fatigue sample was first cut transversely, and then molded into a conductive carbon resin. Before measuring the microhardness, the molded specimens were mechanically polished in two steps. The specimens were first ground using 400, 800, 1200 and 2400 SiC grit paper, successively. They were then, successively, polished using 6, 3, 1 µm diamond polishing paste until a mirror-like finish was obtained.

As for the microhardness measurements, a number of distances below the treated surface were chosen for indentations. The measurements at different distances allow the highlighting of the microhardness variation and, therefore, the gradient properties generated by SMAT, as a function of depth. To minimize the measurement error, for each distance, at least five indentations were carried out. Vickers–hardness indentations were performed using an applied load of 25 g.

### 2.5. Fatigue Tests

The fatigue samples were tested using an ElectroPuls fatigue machine (fabricated by Instron, Norwood, USA). The fatigue tests were carried out at room temperature under torque control using a sinusoidal waveform with a constant amplitude. For the torsional fatigue tests, the nominal shear stress is calculated as the maximum shear stress of the given sample's surface. The expression is as follows [7,35]:

$$\tau = T_r/I_p = 16T/\pi d^3 \tag{1}$$

where $T$ is the applied torque amplitude and $d$ is the central diameter of the sample. For these torsional fatigue tests, the loading frequency is $f$ = 4 Hz. Fatigue failure is defined as

the complete fracture of the sample. If no failure occurred before $2 \times 10^6$ cycles, the fatigue tests were stopped.

To analyze the fracture mechanism due to fatigue, the failed samples were observed using a digital optical microscope (OM, Keyence, Osaka, Japan) and a scanning electron microscope (SEM, Hitachi, Tokyo, Japan).

## 3. Experimental Results

### 3.1. Microhardness Gradient

The in-depth variations of microhardness from the top surface to the center of AM, AM-SMAT, HT and HT-SMAT samples are presented in Figure 2. Note that in the case of the AM state, since samples used in our work were machined by turning, the hardness in the near surface region is slightly higher compared with that of the interior region. The microhardness in the near surface region of the samples after SMAT is obviously increased in comparison with that of the non-SMATed samples for both AM and HT conditions. For AM-SMAT, the maximum hardness value is 187 $HV_{0.025}$ at the treated surface. With the increase in distance, the hardness value decreases gradually to 174 $HV_{0.025}$ at a depth of about 800 µm. This is a common phenomenon in SMATed materials [27], and the increase in hardness in the near surface region is due to a grain refinement combined with a strain-hardening phenomenon, according to the literature [45,46]. It can also be seen that at around 800 µm below the surface, the two curves obtained with SMATed and non-SMATed samples join each other. This means that 800 µm roughly corresponds to the maximum depth affected by SMAT. In addition, the in-depth progressive variation of hardness indicates that there is the presence of gradient features in the SMATed affected region.

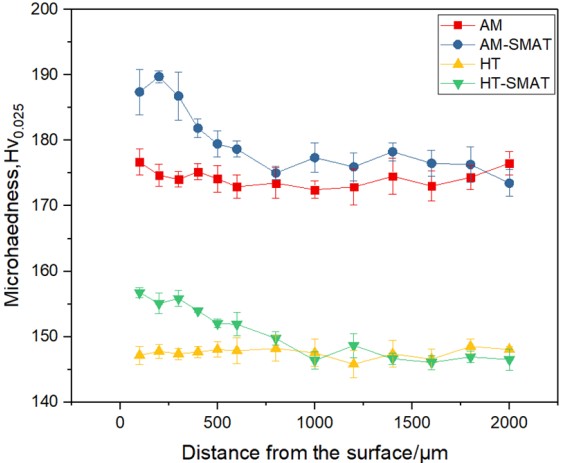

**Figure 2.** In-depth variations of microhardness for samples prepared with different processes.

As for the heat-treated samples (HT and HT-SMAT), the results exhibit a similar trend, but with significantly lower microhardness values for the HT condition. In the center of the heat-treated samples (see the green and yellow curves), the microhardness is around 147 $HV_{0.025}$. Due to the SMAT effect, the microhardness of HT-SMAT is increased to 157 $HV_{0.025}$ at the surface. It should be noticed that the hardness decrease rate shown by the green curve (HT-SMAT) is slightly lower than that of the blue curve (AM-SMAT). This seems to imply that the gradient feature obtained by HT-SMAT is less pronounced than that obtained by AM-SMAT.

### 3.2. Fatigue Tests Results

#### 3.2.1. S–N Curves

Figure 3a compares the S–N curves of the AM and AM-SMAT samples in dual-logarithm scales. To better visualize the effect of SMAT, the obtained data points were fitted

by lines using the Basquin equation. According to the two lines, there is an intersection between them at about 192 MPa. When the applied stress amplitude is higher than 192 MPa, the number of cycles to failure $N$ is higher for AM-SMAT. This means that in this range of stress amplitude, SMAT increases the fatigue resistance, thereby demonstrating a beneficial effect. On the contrary, for the stress amplitudes lower than 192 MPa, the effect seems to be opposite. AM-SMAT samples exhibit lower fatigue strength compared with AM samples in this range, which means that SMAT introduces negative effects to the as-machined alloy.

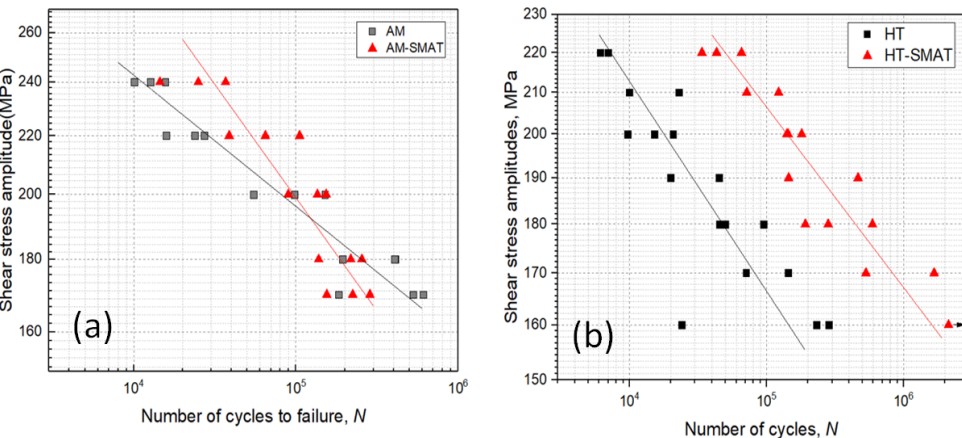

**Figure 3.** S–N plots of experimental torsional fatigue data for (**a**) AM and AM-SMAT samples, and (**b**) HT and HT-SMAT samples. S–N curves built from the Basquin equation (black and red lines) are also plotted.

In the same way, torsional fatigue data for HT material states are presented in Figure 3b. Different from the results of AM and AM-SMAT conditions, the HT-SMAT samples exhibit an overall superior fatigue resistance compared with the HT samples under all the investigated stress levels. Note that a run-out was obtained for the HT-SMAT samples under a stress amplitude of 160 MPa (indicated by an arrow). The more pronounced SMAT effect at lower stress amplitudes seems to be approved by the Basquin lines fitted from the fatigue data, i.e., the improvement in fatigue lives progressively increases with a decrease in stress level.

To model the fatigue data trend and better visualize the SMAT effect, quantitative analysis of the Basquin equation is given below. The Basquin equation used above has the form as follows:

$$\sigma_a = \sigma_f'(2N)^b \tag{2}$$

where $\sigma_f'$ is the fatigue strength coefficient, and $b$ is the fatigue strength exponent. Both $\sigma_f'$ and $b$ can be fitted from the obtained experimental data points. Therefore, the Basquin equations of the torsional fatigue data obtained for different material states are shown below:

$$\text{AM}: \ \sigma_a = 606.47 \times (2N)^{-0.092} \tag{3}$$

$$\text{AM} - \text{SMAT}: \ \sigma_a = 1396.69 \times (2N)^{-0.160} \tag{4}$$

$$\text{HT}: \ \sigma_a = 534.56 \times (2N)^{-0.095} \tag{5}$$

$$\text{HT} - \text{SMAT}: \ \sigma_a = 640.62 \times (2N)^{-0.093} \tag{6}$$

In order to facilitate the interpretation of the results based on the parameters of the Basquin equations, the values of the two parameters $\sigma_f'$ and $b$ for different material states are listed in Table 2. It can be seen from Table 2 that the material treated by SMAT (AM-SMAT and HT-SMAT) has higher $\sigma_f'$ for both AM and HT states. However, the variation of $b$ for the two conditions is different. For the cases of AM and AM-SMAT samples, although the

$\sigma'_f$ value of AM-SMAT increases from 606.46 to 1396.69 MPa, $b$ decreases from $-0.092$ to $-0.160$ at the same time. The lower fatigue lives obtained in the range of low stress level reflect the limited effect of SMAT, which is manifested by the decrease in $b$; in contrast, for the HT and HT-SMAT samples, both two parameters increase from 534.56 to 640.62 MPa for $\sigma'_f$ and from $-0.095$ to $-0.093$ for $b$, respectively, due to the SMAT effect. This means that for the whole stress range studied in this work, there is an improvement in fatigue resistance due to SMAT, and the improvement is more pronounced in the range of low stress amplitude.

**Table 2.** Values of Basquin parameters $\sigma'_f$ and $b$ for different material states.

| Parameter<br>Material State | $\sigma'_f$ | $b$ |
|---|---|---|
| AM | 606.47 | $-0.092$ |
| AM-SMAT | 1396.69 | $-0.160$ |
| HT | 534.56 | $-0.095$ |
| HT-SMAT | 640.62 | $-0.093$ |

3.2.2. Macro-Fracture Analyses

The overall views of the fracture surfaces for AM and AM-SMAT samples obtained under both high (240 MPa) and low (170 MPa) stress levels are given in Figure 4. Note that all the fracture surfaces obtained for these samples are quite flat and perpendicular to the loading axis, according to their lateral fracture profiles observed using a digital optical microscope. Firstly, it can be observed that all the fracture surfaces of both AM and AM-SMAT samples have visible conchoidal stripes and easily identified final rupture regions (center of the stripes). In particular, a murky grey region can be observed in the opposite side of the final rupture region in the AM samples under all the stress amplitudes, as shown in Figure 4a,b. This phenomenon is similar to the results reported by Zhang et al. [7], who indicated that the formation of the murky grey region is due to reversed torsion loading and corresponds to the crack initiation region. However, the murky grey features are not obviously present for the AM-SMAT samples (see Figure 4c,d). Unfortunately, the reason related to the absence of murky grey region of AM-SMAT samples is not clear so far, and requires further investigation.

Similar observations were performed for HT and HT-SMAT samples after fatigue tests. The crack profiles of the samples HT under different torsional loading are shown in Figure 5. Contrary to the case of the AM samples, it can be observed that the fracture appearances are totally different from each other under each stress amplitude. When the stress amplitude is high (200 MPa), the fracture surface has many ridges and valleys, termed 'factory-roof' [47], as illustrated in Figure 5d.

The occurrence of this feature could be due to the fact that the Mode I crack grows faster than the Mode III crack, and the fracture dominated by Mode I cracks leads to the formation of the 'factory-roof' fracture pattern [6]. The fracture surfaces tend to become flat as the shear stress amplitude decreases, as seen Figure 5a–c. The overall fracture surface is almost perpendicular to the sample axis at the low shear-stress amplitude of 160 MPa, according to Figure 5c. Contrary to that obtained under high shear-stress amplitude which shows 'factory-roof' feature, the fracture surface shown in Figure 5c seems to be quite smooth with the presence of visible conchoidal stripes (Figure 5f). This means that the fracture surface is dominated by Mode III cracking under low stress amplitudes [24].

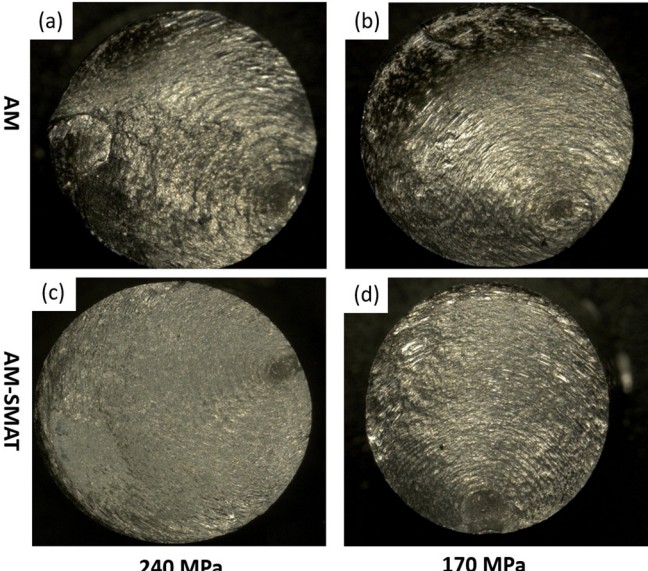

**Figure 4.** Macro-failure analyses of fracture surfaces: (**a**,**b**) correspond to AM state under 240 MPa and 170 MPa, respectively; (**c**,**d**) correspond to AM-SMAT state under 240 MPa and 170 MPa, respectively. Note that the diameter of the fracture surfaces is about 4 mm.

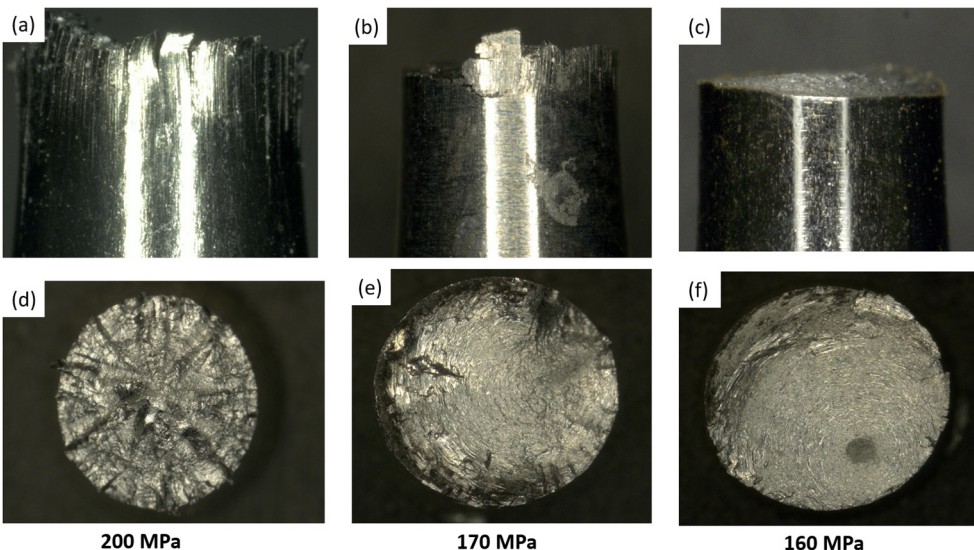

**Figure 5.** Macro-failure analysis of HT samples: (**a**,**d**) correspond to the shear stress amplitude of 200 MPa, (**b**,**e**) correspond to 170 MPa, and (**c**,**f**) correspond to 160 MPa. Note that the diameter of the fracture surfaces is about 4 mm.

It should be noted that the results presented here are similar to those obtained in the work of Li et al. [6]. They found that the fracture morphologies of coarse-grained copper during torsion fatigue exhibit features similar to those in our work, i.e., 'factory-roof' exists for higher strain amplitudes and smooth fracture surfaces are observed for lower strain amplitudes. However, the results presented in other studies in the literature are different [47,48]. In these reports, the fracture surfaces of steel and titanium change from a 'factory roof' pattern to a flat feature as the shear stress amplitude increases. This phenomenon may be due to the fact that the grain structure characteristic of the face-centered cubic (fcc) metals, such as copper and aluminum, is different from that of hexagonal close-packed (hcp) metals, such as titanium [49].

### 3.2.3. Micro-Fracture Analyses

SEM observations of representative HT samples tested under high (210 MPa) and low (170 MPa) stress levels were carried out, as shown in Figure 6. Under 210 MPa, longitudinal and circumferential cracks encounter and connect with each other, constituting a large damage zone on the circumferential surface of samples (Figure 6a). According to the literature [6,24], longitudinal cracks occur first in Mode II in the case of torsional fatigue tests. When the longitudinal cracks grow to a critical length and penetrate inwards, the stress concentration at the tips of these cracks will give rise to the occurrence of circumferential cracks. Many Mode I cracks turn out along the radial direction on the cross-section in the case of high shear stress amplitude, as shown in Figure 6b. The appearance mentioned here is 'factory-roof' as presented above. The fatigue striations accompanying the macro-cracks can be observed on the fracture surface, as shown in Figure 6c.

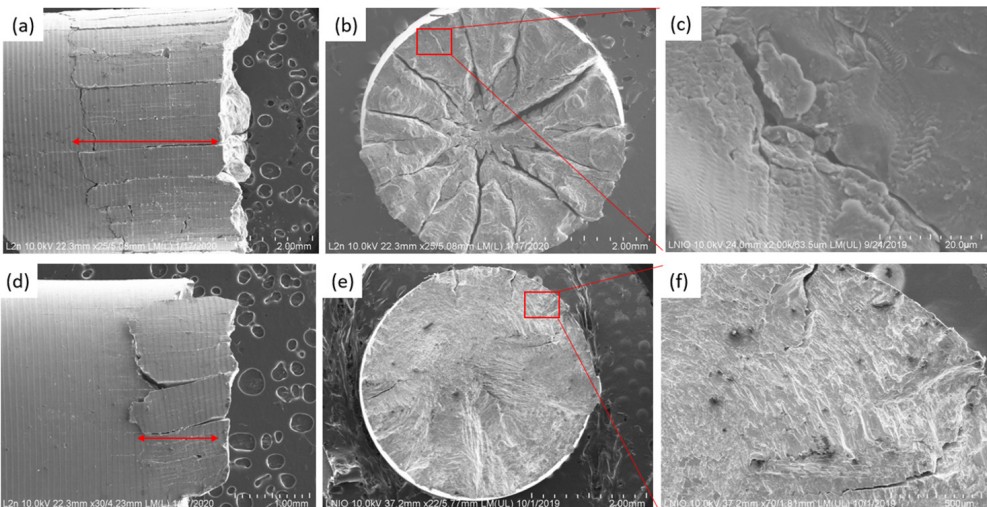

**Figure 6.** Fracture morphologies of HT samples: (**a–c**) correspond to 210 MPa, and (**d–f**) correspond to 170 MPa. Among these figures, circumferential surfaces (**a**,**d**), fracture surfaces (**b**,**e**), and magnified views (**c**,**f**) of red rectangles shown in (**b**,**e**) are shown, respectively.

The observation of the failed HT sample tested under low shear-stress amplitude (170 MPa) is given in Figure 6d. Compared with that of the high stress level, displayed in Figure 6a, the number and the length of longitudinal cracks (Mode II) are lower for HT samples under a low stress level. Consequently, the damage zone obtained under the low stress level is smaller than that obtained under the high stress level. The fracture surface of the HT sample obtained under 170 MPa is given in Figure 6e. It seems to be smoother than that of the sample tested under the high stress level, according to the comparison between Figures 6b,e. In a magnified image of the fracture surface shown in Figure 6f, several cracks along the radial direction (Mode I) can also be observed. However, the fatigue striations can hardly be observed and a dominant part of the fracture surface is occupied by ring-like features.

Similar observations were carried out for HT-SMAT samples tested under high (220 MPa) and low (170 MPa) stress levels; several representative examples are shown in Figure 7. Similar to the HT cases, both longitudinal and circumferential cracks can be observed on the lateral surface under the stress amplitude of 220 MPa (Figure 7a). However, these longitudinal cracks are shallower and narrower than those in the HT sample. The fracture surface of HT-SMAT sample under 220 MPa is 'factory roof'-like, as seen in Figure 7b, which is similar to the fracture surface of the HT sample under the high stress level (see Figure 6b). According to the magnified view of the 'factory roof' pattern as displayed in Figure 7c, many fatigue striations can be observed.

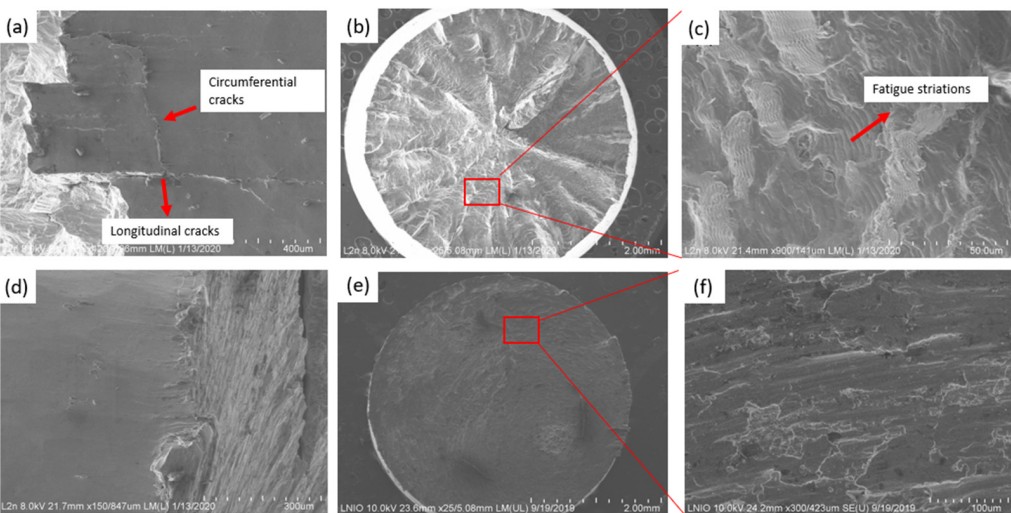

**Figure 7.** Fracture morphologies of HT-SMAT samples: (**a**–**c**) correspond to 220 MPa and (**d**–**f**) correspond to 170 MPa. Among these figures, circumferential surfaces (**a**,**d**), fracture surfaces (**b**,**e**), and magnified views (**c**,**f**) of red rectangles shown in (**b**,**e**) are shown, respectively.

In the case of HT-SMAT sample under the low shear-stress level, however, only a few longitudinal cracks (Mode II) can be observed, as shown in Figure 7d. Compared with the HT sample under the same stress level (see Figure 6e), a few cracks on the lateral surface result in a smoother fracture surface for the HT-SMAT sample, as displayed in Figure 7e. Hence, it could be concluded that the fracture surface changes from 'factory roof' to a flat feature with the decrease in stress level for both HT and HT-SMAT states. The magnified view shown in Figure 7f indicates that the ring-like features instead of fatigue striations occupy a dominant part of the fracture surface.

## 4. Analysis and Discussion

### 4.1. Fracture Mechanism

The process of torsional fatigue failure may be divided into different stages. For ductile materials, one of the most popular descriptions is the crack propagation model with three stages, presented by Forsyth et al. [50]. Under torsional loading, shear stress causes fatigue damage and is regarded as the driving force for fatigue crack initiation. Once the fatigue cracks initiate on the maximum shear stress planes, the maximum principal stress will play a dominant role in the crack transformation from Stage I to Stage II, thereby promoting the propagation of fatigue cracks [49]. Higher principal stress can lead to the occurrence of the Stage II crack propagation, and thus the fracture surfaces are different under different stress amplitudes [51].

In both HT and HT-SMAT samples under high shear-stress levels, longitudinal cracks constitute a large damage zone at the circumferential surface, as illustrated in Figures 6a and 7a. These cracks then propagate along the oblique direction and lead to the fracture of samples with the typical character of Stage II, i.e., fatigue striations, as displayed in Figures 6c and 7c. In other words, longitudinal cracks form and propagate first in Stage I, and then change to Stage II with a direction of 45° to the longitudinal axis due to the promoting effect of maximum principal stress. Hence, under high stress levels, longitudinal cracks grow longer and result in a wider damage zone. Within the damage zone, the cracks propagate along the oblique direction in Mode I, which generates fatigue striations on the fracture surface. Note that during this process, Mode I cracks tend to grow faster than Mode III cracks, which gives rise to the formation of the 'factory-roof' pattern. Therefore, it is the wider damage zone and the larger driving force that make the Stage II crack propagation occur.

For the HT and HT-SMAT samples under low stress levels, longitudinal cracks can also constitute a damage zone on the circumferential surface, as seen in Figures 6d and 7d.

However, the damage zones obtained under low stress levels are significantly smaller than those obtained under high stress levels. Moreover, under low stress amplitudes, the cracks seem to keep their initial directions and do not switch to Stage II propagation, due to the lack of sufficient maximum principal stress, which determines the changing of crack direction. Therefore, cracks are prone to only grow in Stage I and it is hard for Stage II crack propagation to occur. This means that the final fracture surfaces do not show fatigue striation markings but only ring-like features, as seen in Figures 6f and 7f.

In the case of the AM and AM-SMAT samples, the number of longitudinal cracks decreased with the decrease in stress level, which is similar to the cases of the HT and HT-SMAT samples. Simultaneously, the width of the cracks also decreased with the decrease in stress level. However, all the fracture surfaces of AM samples are quite 'flat' and almost perpendicular to the samples' axis for all the stress amplitudes investigated in this work. This observation is consistent with the results presented by Zhang et al. [35]. Furthermore, only ring-like features instead of fatigue striations can be observed on the fracture surfaces under all the stress levels, according to the fracture surface observations. This seems to indicate that shear stress dominates the fracture process for AM and AM-SMAT samples under all the stress amplitudes. In addition, it implies that the cracks propagate only in Stage I and their directions do not change to Stage II, even under high stress levels.

### 4.2. Effect of SMAT on Torsional Fatigue Properties

SMAT improves the torsional fatigue resistance for HT samples under all the stress levels and for the AM samples under high stress levels. Unfortunately, the SMAT effect is not obvious under low stress levels for the AM samples. Both these beneficial and detrimental effects of SMAT on torsional fatigue properties should be understood.

SMAT could effectively limit the initiation of longitudinal cracks on the lateral surface of a sample, according to the comparison of lateral surface observations between HT and HT-SMAT samples under high and low stress levels, as shown in Figures 6 and 7. The Mode II cracks of HT samples under the high stress level of 210 MPa are much larger and obvious as seen in Figure 6a. In the case of HT-SMAT samples shown in Figure 7a, it can be clearly seen that the cracks are shallower and narrower than those of the HT sample under the stress level of 210 MPa. A similar phenomenon can be observed in the case of the low stress level of 170 MPa, based on the comparison between Figures 6d and 7d. Remember that all the HT-SMAT samples have higher fatigue lives than those of HT under the same stress level. Therefore, it could be concluded that the effect of SMAT leads to a smaller damage zone and, accordingly, results in higher fatigue strength.

A similar result was reported in the work of Wang et al. [24], who suggested that gradient microstructure was generated in pure titanium (Ti) by means of surface rolling treatment (SRT). In their work, the presence of a stress gradient is considered to delay the initiation of cracks and lead to the enhanced fatigue life of Ti. Li et al. [6] reported that the stress gradient of solid cylinder samples subjected to cyclic torsional loading has an influence on the crack propagation behavior. Therefore, in our work, the effect of the stress gradient introduced by SMAT on the fatigue behavior can be regarded as the main beneficial effect due to SMAT.

It is known that torsional fatigue cracks usually grow on the surface of samples, obeying the direction given by Mode II + III. The growth of longitudinal cracks along the length and the depth directions is controlled, respectively, by pure Mode II and Mode III, as shown in Figure 8. It is assumed that the longitudinal cracks are semi-elliptical. The front edge of the major axis of the semi-elliptical crack is under the condition of Mode II and the front edge of the minor axis is under Mode III. Therefore, the stress intensity factors $K_{II}$ and $K_{III}$ exist along the semi-elliptical crack fronts, as indicated in Figure 8.

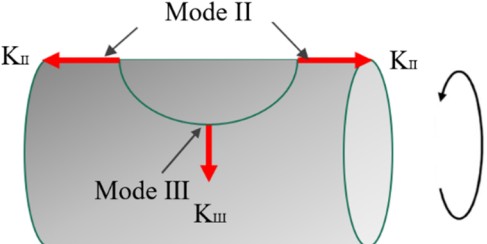

**Figure 8.** The modes of longitudinal crack propagation under torsional loading.

Murakami et al. [52,53] suggested that $K_{II}$ is essentially equal to $K_{III}$, which means that a 3D longitudinal crack grows under the condition of $K_{II}=K_{III}$. According to Wang et al. [24], the following relation can be obtained:

$$\frac{\tau_{II}}{\tau_{III}} \cdot \frac{1}{1-\nu} \cdot \left(\frac{b}{a}\right)^{1/2} = 1 \tag{7}$$

where $a$ is half of crack length, $b$ is crack depth, and $0 < b/a < 1$. $\tau_{II}$ and $\tau_{III}$ are the shear stresses at the tip of the longitudinal crack along the length and the depth directions, respectively. $\nu$ is Poisson's ratio of material. $\frac{\tau_{II}}{\tau_{III}}$ can be used to represent the stress gradient level, and the value of $\frac{b}{a}$ determines the shape of the crack. It can be seen from Equation (7) that the relationship between the stress gradient level $\frac{\tau_{II}}{\tau_{III}}$ and the shape of the crack represented by $\frac{b}{a}$ is inverse, i.e., as the stress gradient level increases, the size of the longitudinal crack sharply decreases [24].

Several factors introduced by SMAT in this work probably influence the stress gradient. According to the work of Wang et al. [24,54], the stress gradient level in the surface nanocrystallized (SNC) Ti treated by SRT is higher than that in the non-treated Ti, due to the effect of compressive residual stress [31]. Besides the effect of compressive residual stress, grain refinement is another factor which can affect the stress gradient level according to Li et al. [31]. In their work, in order to facilitate the comparison, cases with applied stress amplitudes near the fatigue limit were considered. They found that the stress gradient level of ultra-fine grained (UFG) Cu was larger than that of coarse-grained (CG) Cu near the surface of the sample.

Consequently, in our work the stress gradient level of the samples treated by SMAT (AM-SMAT and HT-SMAT) should be higher than the untreated samples (AM and HT), due to the compressive residual stress and the grain refinement introduced by SMAT. Hence, it is not difficult to conclude that the increased stress gradient level of HT-SMAT will limit crack initiation and, therefore, prolong the fatigue lives of samples under each stress amplitude. However, in the case of AM-SMAT, SMAT led to a decreased fatigue resistance under low stress levels, as mentioned above. The detrimental effect of SMAT resulted in a decrease in fatigue lives in the low stress range, as seen in Figure 3a. In general, the rough surface introduced by SMAT decreases fatigue life because it tends to induce the initiation of cracks [55]. Rough surface is a negative factor, as it generally causes local stress concentration that promotes crack nucleation from the surface, thereby degrading the fatigue performance of materials [56–58]. According to Maurel et al. [3], the high notch sensitivity of AA7075 alloy could explain why SMAT cannot improve its high cycle fatigue resistance, compared to other aluminum alloys such as AA2024. Moreover, according to Akiniwa et al. [28], the fatigue strength of harder materials is very sensitive to defects. It is well known that a rough surface can provide defects which are detrimental to fatigue strength [29]. Note that the hardness value of AM-SMAT is the highest among all the states (see Figure 2). AM-SMAT samples, thus, show lower fatigue strength than AM under low stress levels. This could be due to the fact that the fatigue strength of AM-SMAT is more sensitive to roughness than other states. Furthermore, it is known that fatigue life is dominated by crack initiation under a low stress level [30], and surface roughness



has a greater effect on fatigue initiation [59]. A similar phenomenon was obtained by Rai et al. [30], who found a degraded fatigue life of a stainless steel treated by Ultrasonic Shot Peening (USSP) under low strain amplitudes. In their work, the authors claim that the surface cracks induced by USSP increased the average surface roughness and resulted in reduced fatigue life.

In addition, it is widely recognized that grain refinement has an influence on both crack initiation and propagation [31]. On the one hand, grain refinement could increase the resistance to fatigue crack initiation owing to strength enhancement. The resistance to crack growth, on the other hand, is known to decrease with a decrease in grain size, since a less tortuous crack path is generated during fatigue crack growth [60]. The grain refinement near the surface of AA7075 alloy resulting from SMAT could accelerate the crack propagation and thus result in poor fatigue performance under low stress levels. In the work conducted by Li et al. [31], they found a similar phenomenon through studying ultrafine-grained and coarse-grained copper. As a matter of fact, the decreased fatigue resistance implies that there is a competition between the detrimental and the beneficial effects of different factors introduced by SMAT, and the detrimental effects are dominant for AM-SMAT samples under low stress amplitudes.

## 5. Conclusions

The fatigue properties of an AA7075 alloy in AM and HT states treated by SMAT were investigated under torsional loading. The failed samples were observed using digital optical microscopy and SEM in order to analyze the mechanisms related to the SMAT effects under different stress levels. Several main conclusions can be drawn:

1. AM-SMAT samples show superior fatigue strength only in the range of high stress levels. The detrimental effect of SMAT leads to the reduction in fatigue life under low stress levels. HT-SMAT samples have markedly higher fatigue lives under all the stress levels investigated in this work compared with HT samples, according to the analyses of the S–N data.
2. The fracture mechanism of torsional fatigue tests can be described as follows: the micro-cracks initiate and propagate on the circumferential surface along maximum shear planes (either parallel or perpendicular to the axis of the samples), which is known as Stage I. In Stage II, the cracks change their directions to propagate along maximum principal stress planes (45° to the axis), due to the effect of principal stress.
3. SMAT leads to a narrow damage zone on the lateral surface of sample and, therefore, results in a decreased number of longitudinal cracks. This phenomenon is related to the higher stress gradient level introduced by SMAT, which can hinder the initiation and propagation of longitudinal cracks and lead to increased fatigue lives. However, the rough surface and the acceleration of crack propagation, due to grain refinement, are regarded as detrimental factors, which lead to decreased fatigue lives in the low stress range.

**Author Contributions:** Conceptualization, Z.S. and D.R.; methodology, Y.L. and P.G.; validation, Z.S., D.R. and H.X.; formal analysis, Y.L.; investigation, Y.L. and P.G.; data curation, Y.L.; writing—original draft preparation, Y.L; writing—review and editing, T.G. and Z.S.; supervision, Z.S. and D.R.; project administration, D.R. All authors have read and agreed to the published version of the manuscript.

**Funding:** This research received no external funding.

**Institutional Review Board Statement:** Not applicable.

**Data Availability Statement:** Not applicable.

**Acknowledgments:** Y.L. would like to express his cordial gratitude to the Chinese Scholarship Council (CSC) for financially supporting his Ph.D. study.

**Conflicts of Interest:** The authors declare no conflict of interest.

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
