# Peer review of "Effect of Surface Mechanical Attrition Treatment on Torsional Fatigue Properties of a 7075 Aluminum Alloy"

_metals, doi:10.3390/met12050785_

Round 1
Reviewer 1 Report
Very interesting paper!
The concept of using SMAT methods to increase the resulting properties of machined parts is very elegant and the paper proves the concept right.
Still some minor revision is needed:
- The Fig. 4 is obviously not depicted at all! There are only some dots and no line whatsoever.
- This leads to some failures in the text in lines 203, 207 and 249.
- It is obvious that the method of machining doesn’t affect the result BUT! It still would be beneficial for the scientific part of the work if the machining method would be named. If was machined by electro-erosion, then this method leaves the least surface deformation after cutting and the surface modification effect is only from SMAT. BUT if the samples were machined by turning, then it is obvious that a part of the surface hardening is attributable to the deformation after turning. In that case it would be nice to evaluate the impact of both the turning and the SMAT in the resulting effect. The effect of the SMAT is stronger obviously, but for the purpose of scientific clarity it will be better.
Best regards!
Author Response
Dear editor and reviewers,
We really appreciate that you took the time to review this manuscript. Your comments and suggestions helped us catch some of the errors that we missed and provided valuable inputs to improve this manuscript. After reviewing all the comments and suggestions, we have revised the manuscript accordingly, and the responses to the comments are listed hereafter.
Response to Reviewer 1:
Comment 1: The Fig. 4 is obviously not depicted at all! There are only some dots and no line whatsoever.
Response: The Fig. 4 aims to explain the effect of SMAT through the two parameters namely fatigue strength coefficient and fatigue strength exponent b for different material states. The values of these two parameters are determined from the Basquin equations fitted based on the experimental data (see Eq. (3)-(6)). Therefore, these dots are independent of each other and there is no need to add lines to connect these points.
However, by taking into account this comment, we think that it could be better to change this figure into a table (see Table 2 in the manuscript). Consequently, the sentences that describe these two parameters have been adapted to this change.
Comment 2: This leads to some failures in the text in lines 203, 207 and 249.
Response: We are very sorry that some errors occurred when the document was transformed to PDF version. We have therefore corrected them in the manuscript.
Comment 3: It is obvious that the method of machining doesn’t affect the result BUT! It still would be beneficial for the scientific part of the work if the machining method would be named. If was machined by electro-erosion, then this method leaves the least surface deformation after cutting and the surface modification effect is only from SMAT. BUT if the samples were machined by turning, then it is obvious that a part of the surface hardening is attributable to the deformation after turning. In that case it would be nice to evaluate the impact of both the turning and the SMAT in the resulting effect. The effect of the SMAT is stronger obviously, but for the purpose of scientific clarity it will be better.
Response: The samples used in our work were machined by turning. This machining process may lead to an increase in hardness in the near surface region of the samples. We have added some information about this point in Section 3.1.

Reviewer 2 Report
Congratulations to the authors for doing this practical work. They tried to study the simultanous effects of heat treatment and ultrasound shot peening using steel balls in the enclosed chamber on the torsional fatigue behavior of Al-7075-T6. Also, they investigated the macro and micro failure analysis and other one. In conclusion, the presented results are of great value. but, the authors should attention to the following points and I believe that some corrections are need to publish this manuscript.
1- By performing surface treatment with the aim of fatigue improvement such as SMAT process, two parameters of grain size and compressive residual stress affected on the fatigue behavior of the metallic materials. So, it is strongly recommended to measure grain size in the surface and depth of all specimen batches. In other words, present the diagram of grain size in terms of depth. In addition, it is necessary to measure the residal stress due to SMAT process in both surface and depth.
2- To interpret the usefulness and harmfulness of SMAT process on samples without heat treatment, it is better to use the results of grain size, hardness, microscopic observations, residual stress, and fatigue data Hardness in one time and describe the reason for this event.
3- Authors are advised to follow the experiments in the same articles ("https://doi.org/10.1016/j.surfcoat.2018.02.081", "https://doi.org/10.1016/j.ijfatigue.2018.06.004", and "https://doi.org/10.1016/j.matchar.2019.109877"). In addition, These documents can also be used to complete the literature review section.
4- In the section 2.1 "Material", it is better to state the used standard for fabricate fatigue testing specimens and used standard to perform torsional fatigue test.
5- It is better to report the surface details of fatigue specimens such as surface roughness.
6- The phrase "... is shown in Table 1" should change to ".... is reported ..." or "... is given ..."
7- Related to the section 2.3 "SMAT process", it is necessary to how a schematic of the SMAT process. It is also good to show a picture of the device used for this process. Moreover, describe more details in this process such as frequency and amplitudes of excitation vibration related to the steel balls.
8- The authors stated that the fatigue samples rotate during the SMAT process, so please report the rotating speed.
9- In the process of SMAT, how do you know that the entire surface of the middle part of sample hited by steel balls? How long does this process take? How many balls are used to perform the process.
10-There are errors in referring reference (see lines 203 and 207).
Author Response
Dear editor and reviewers,
We really appreciate that you took the time to review this manuscript. Your comments and suggestions helped us catch some of the errors that we missed and provided valuable inputs to improve this manuscript. After reviewing all the comments and suggestions, we have revised the manuscript accordingly, and the responses to the comments are listed hereafter.
Response to Reviewer 2:
Comment 1: By performing surface treatment with the aim of fatigue improvement such as SMAT process, two parameters of grain size and compressive residual stress affected on the fatigue behavior of the metallic materials. So, it is strongly recommended to measure grain size in the surface and depth of all specimen batches. In other words, present the diagram of grain size in terms of depth. In addition, it is necessary to measure the residual stress due to SMAT process in both surface and depth.
Response: Thanks for this comment. Unfortunately, we were not able to perform these observations/measurements of grain size and residual stresses due to the Covid-19 restrictions (for several months in 2020) imposed in France during Dr. Li’s PhD study. However, for the residual stress measurement, in our other work conducted in collaboration (T. Gao et al. “Effect of surface mechanical attrition treatment on high cycle and very high cycle fatigue of a 7075-T6 aluminium alloy”, Int. J. Fatigue, 2020, 139, 105798), very similar SMAT conditions were used to treat the same 7075-T6 aluminum alloy. In that work, we investigated the in-depth residual stress distribution, as seen in the figure below. It is demonstrated that SMAT is capable of generating a compressive residual stress field in the region close to the surface of a structure with similar dimensions to ours. As for the grain size distribution, similar treatments were performed in the literature for various materials including aluminum alloys, and grain size gradient has been highlighted by many researchers.
Figure showing residual stress distribution extracted from our previous work: T. Gao et al., Effect of surface mechanical attrition treatment on high cycle and very high cycle fatigue of a 7075-T6 aluminium alloy, Int. J. Fatigue, 2020, 139, 105798.
Even though the observations/measurements for these two aspects have not yet been performed, we think that with the results commonly presented in the literature, it would be possible to qualitatively interpret the results obtained in this work, as we have done in this paper.
However, based on this comment, we added a sentence with some references in Section 2.3 to support our interpretations of results with the involvement of grain size and residual stress distributions. This sentence is the following:
According to the literature for various SMATed materials, this treatment should be able to generate a grain size gradient with a superficial nanostructured layer at the treated surface [45,46,47] as well as a compressive residual stress field [18].
Comment 2: To interpret the usefulness and harmfulness of SMAT process on samples without heat treatment, it is better to use the results of grain size, hardness, microscopic observations, residual stress, and fatigue data Hardness in one time and describe the reason for this event.
Response: We totally agree with the reviewer that to get a deep understanding in a quantitative manner, a systematic study of all these mentioned parameters is necessary. This actually constitutes our future work that is ongoing with another PhD student. However, this article aims to comparatively investigate the fatigue properties of the aluminum alloy under different states. With the effects of SMAT commonly reported in the literature, it would be possible to quantitatively interpret the results obtained in this work.
Comment 3: Authors are advised to follow the experiments in the same articles ("https://doi.org/10.1016/j.surfcoat.2018.02.081","https://doi.org/10.1016/j.ijfatigue.2018.06.004", and "https://doi.org/10.1016/j.matchar.2019.109877"). In addition, these documents can also be used to complete the literature review section.
Response: Thanks for your advice. We have added these references in the literature review section. For our future work concerning the detailed investigation of the different SMAT induced parameters, we will refer to the experimental methods used in these articles.
Comment 4: In the section 2.1 "Material", it is better to state the used standard for fabricate fatigue testing specimens and used standard to perform torsional fatigue test.
Response: In our work, to design the fatigue specimen by taking into account different factors (possibility of applying SMAT, capacity of the machine…), we referred to the literature (for example the work of J. Zhang et al. “Effect of mean shear stress on torsion fatigue failure behavior of 2A12-T4 aluminum alloy”, and J. Zhang et al. “Tension-torsion high-cycle fatigue failure analysis of 2A12-T4 aluminum alloy with different stress ratios”). In the standard dedicated to torsional fatigue tests, a tubular specimen is usually used. In our work, as the fatigue specimens would be processed by SMAT, the specimens should be strong enough to undergo impact loading. We thus decided to adopt the currently used specimen.
A sentence was added in Section 2.1 to clarify this point:
The dimensions and the shape of the samples (see Fig. 1) were determined by referring to the literature [7,37,38] and by taking into account other factors (possibility of being treated by SMAT, capacity of the fatigue machine).
Comment 5: It is better to report the surface details of fatigue specimens such as surface roughness.
Response: Actually, we indeed assessed the roughness of the specimens with different surface states, as shown in two figures below.
Surface observation of specimens in different states: (a) As-machined (AM), (b) AM-SMAT.
Surface profiles measured along the axial direction of specimens in different states: (a) As-machined (AM), (b) AM-SMAT.
Based on this comment, some representative roughness values were added in the manuscript:
Note that the roughness of the samples at these states was measured and some representative values are presented as follows: arithmetic mean roughness Ra = 0.45 µm and total height of the roughness profile Rt = 4.75 µm for non-SMATed samples (AM and HT), and Ra = 0.25 µm and Rt = 6.17 µm for SMATed ones (AM-SMAT and HT-SMAT).
Comment 6: The phrase "... is shown in Table 1" should change to ".... is reported ..." or "... is given ..."
Response: “…is shown in Table 1…” has been changed into “is given”.
Comment 7: Related to the section 2.3 "SMAT process", it is necessary to show a schematic of the SMAT process. It is also good to show a picture of the device used for this process. Moreover, describe more details in this process such as frequency and amplitudes of excitation vibration related to the steel balls.
Response: A schematic of a SMAT set-up can be rather easily found in the literature. This is why we did not include it in our manuscript. According to this comment, a sentence and two references were added in Section 2.3. In addition, more details about the treatment conditions (frequency, amplitude of the vibration…) were given in the same section. A picture of a specimen installed in the SMAT device is shown below:
Illustration of a specimen installed in the SMAT device for a treatment.
Comment 8: The authors stated that the fatigue samples rotate during the SMAT process, so please report the rotating speed.
Response: The rotating speed used in our work is 30 rpm. We have added this information in the Section 2.3.
Comment 9: In the process of SMAT, how do you know that the entire surface of the middle part of sample hitted by steel balls? How long does this process take? How many balls are used to perform the process.
Response: In this work, to control the coverage applied to the surface of the middle part of specimens impacted by steel balls, we performed coverage assessment. During the coverage assessment, interruptions were done in order to check the specimen surface using an optical microscope. After 150 seconds, it seems that the whole specimen surface is fully covered by ball impacts, since the colour initially marked on the surface of the specimen has almost totally disappeared as seen in the figure below.
Surface topography of a specimen treated by SMAT: (a) before the treatment, (b) treated by SMAT during 150 s
The information about the coverage for the SMATed specimens was added in Section 2.3:
In this work, according to the coverage assessment performed for a sample, a duration of 150 seconds is regarded as 100% coverage of ball impacts. Therefore, a duration of 25 minutes corresponding to a coverage of 1000% was chosen.
Comment 10: There are errors in referring reference (see lines 203 and 207)
Response: Some errors occurred while the document was transformed to PDF version. This type of errors has been corrected. Thank you very much.

Reviewer 3 Report
The article highlights peculiarities of the effect of surface mechanical attrition treatment on microhardness and torsional fatigue properties of a 7075 aluminum alloy. The authors analyzed the fatigue fracture mechanisms for samples after various treatments. In the section "Analysis and discussion", the authors state that the roughness and grain size affect fatigue life. However, they do not present studies of roughness and do not estimate the grain size.
The article is interesting, but a number of shortcomings need to be corrected:
- Please remove the following text (Lines 71-76) from the section “Introduction”: “The experimental procedures are briefly described in Section 2. The results of experimental tests including microhardness, S-N plots, along with fracture analyses at both macroscopic and microscopic scales are presented in Section 3. In this section, the fatigue results are analyzed based on a comparison between different conditions. The effects of SMAT and the cracking mechanisms for samples in different states are then discussed in Section 4.”
- Please substantiate the selection of temperature-time parameters of heat treatment (Section 2.2).
- It is unclear how ??′ and b are defined.
- Please explain the text “…Figure 4Error! Reference source not found..” (Line 203).
- Please explain the text “…Error! Reference source not found.” (Line 207).
- Please explain how macro-failure analyzes of fracture surfaces were made at different magnifications, since in the images they are the same, and the magnifications differ twice (x100 vs. x50).
- 6a, b, c, d, e are of poor quality, so it is impossible to analyze fractographic features based on these figures.
- Please explain the text “…Figure 6Error! Reference source not found.(c)” (Line 249).
- For each alloy variant, the predominant influence of crack growth conditions should be indicated (Mode I or Mode II or Mode III).
- 7 and Fig. 9 should be combined: Fig. 9a duplicates Fig. 7a, and Fig. 9c duplicates Fig.7d. It should also be noted that the same image (Fig. 7a, Fig. 9a) has different captions (Fig. 7a corresponds to 210 MPa, whereas Fig. 9 corresponds to 220 MPa).
- Please clarify how the fatigue tests were conducted. Have the tests been conducted under tension-torsion or torsion loading? The loading conditions must be clearly described in Section 2.5.
- More new References (2018-2022) should be added.
Author Response
Dear editor and reviewers,
We really appreciate that you took the time to review this manuscript. Your comments and suggestions helped us catch some of the errors that we missed and provided valuable inputs to improve this manuscript. After reviewing all the comments and suggestions, we have revised the manuscript accordingly, and the responses to the comments are listed hereafter.
Response to Reviewer 3:
Comment 1: Please remove the following text (Lines 71-76) from the section “Introduction”: “The experimental procedures are briefly described in Section 2. The results of experimental tests including microhardness, S-N plots, along with fracture analyses at both macroscopic and microscopic scales are presented in Section 3. In this section, the fatigue results are analyzed based on a comparison between different conditions. The effects of SMAT and the cracking mechanisms for samples in different states are then discussed in Section 4.”
Response: The sentences mentioned by the reviewer have been removed according to his advice.
Comment 2: Please substantiate the selection of temperature-time parameters of heat treatment (Section 2.2).
Response: Due to the trade-off relationship of strength and elongation, the elongation (that can represents the ductility to some extent) of AA7075 is often limited, which could lead to a poor effect when the material is processed by SMAT. Therefore, based on the literature, to obtain an enhanced ductility of material combined with a quite high mechanical strength, several groups of temperature-time parameters of heat treatment were first selected. Furthermore, a series of tensile tests were performed by comparatively assess the ductility and the mechanical strength of the material obtained after the heat treatments with the above selected temperature-time parameters, as shown in the figure below. The red line corresponds to the heat treatment condition finally chosen in the work.
To clarity this point, we have added several references about the selection of temperature-time parameters of heat treatment, and a relevant sentence in Section 2.2.
Comparative investigation of the tensile behaviour of the AA7075 subjected to different heat treatment conditions.
Comment 3: It is unclear how ??′ and b are defined.
Response: As indicated in the manuscript, these two parameters are linearly fitted based on the experimental data ( versus plot, as indicated in the manuscript). Examples of the obtained lines can be seen in Fig. 3.
A table that lists all the values of these two parameters is built (see Table 2 in the revised manuscript), which could facilitate the comparison between the different material states.
Comment 4: Please explain the text “…Figure 4Error! Reference source not found..” (Line 203).
and Comment 5: Please explain the text “…Error! Reference source not found.” (Line 207).
Response: Very sorry for some errors that occurred when the document was transformed to PDF version. We have corrected them.
Comment 6: Please explain how macro-failure analyzes of fracture surfaces were made at different magnifications, since in the images they are the same, and the magnifications differ twice (x100 vs. x50).
Response: The magnificent scales shown in some previous pictures are wrong. The scales are removed because of their unusefulness, since all the specimens are broken roughly at the center and the fracture surfaces have a very similar dimension (a diameter of 4 mm, as indicated in the captions of Fig. 4 and Fig. 5).
Comment 7: 6a, b, c, d, e are of poor quality, so it is impossible to analyze fractographic features based on these figures.
Response: We have replaced them with a group of pictures of better quality.
Comment 8: Please explain the text “…Figure 6Error! Reference source not found.(c)” (Line 249).
Response: This type of errors has been corrected throughout the paper.
Comment 9: For each alloy variant, the predominant influence of crack growth conditions should be indicated (Mode I or Mode II or Mode III).
Response: We have indicated the corresponding crack modes in some relevant places in the manuscript.
Comment 10: 7 and Fig. 9 should be combined: Fig. 9a duplicates Fig. 7a, and Fig. 9c duplicates Fig.7d. It should also be noted that the same image (Fig. 7a, Fig. 9a) has different captions (Fig. 7a corresponds to 210 MPa, whereas Fig. 9 corresponds to 220 MPa).
Response: While preparing the original version of the manuscript, we actually hesitated about this point. According to this comment, we combined these two figures. As for the stress values (210 MPa and 220 MPa), it was actually a mistake.
Comment 11: Please clarify how the fatigue tests were conducted. Have the tests been conducted under tension-torsion or torsion loading? The loading conditions must be clearly described in Section 2.5.
Response: In this work, a tension-torsion fatigue machine was used, but the fatigue tests were conducted under only torsional loading (with the displacement being free, i.e. the axial load is zero). Therefore, only the torsional stress was applied. We have revised this point in Section 2.5 to avoid confusion.
Comment 12: More new References (2018-2022) should be added.
Response: Thanks for the advice. We have added some new references related to this work.

Round 2
Reviewer 2 Report
The authors tried to respond to the comments completely, but I still believe that to complete this article, it would be better to take measurements of grain size and residual stress at the surface and depth. However, the quality of the revised manuscript is more acceptable than before and can be published in the present form.
Author Response
Comment: The authors tried to respond to the comments completely, but I still believe that to complete this article, it would be better to take measurements of grain size and residual stress at the surface and depth. However, the quality of the revised manuscript is more acceptable than before and can be published in the present form.
Response: Thank you for your approval.
Reviewer 3 Report
The authors took into account the comments of the reviewer and made appropriate corrections to the manuscript.
But the References (4, 5, 8, 9, 10, 11, 12, 13, 14, 15, 16, 17, 20, 21, 25, 26, 31, 32, 33, 34, 36, 44, 50, 51, 52, 53, 54, 57) should be corrected, namely, journal, year, issue and pages should be added.
Author Response
Response to Reviewer 3:
Comment 1: The authors took into account the comments of the reviewer and made appropriate corrections to the manuscript.
But the References (4, 5, 8, 9, 10, 11, 12, 13, 14, 15, 16, 17, 20, 21, 25, 26, 31, 32, 33, 34, 36, 44, 50, 51, 52, 53, 54, 57) should be corrected, namely, journal, year, issue and pages should be added.
Response: The relevant information of References has been added in the manuscript.